# Normalising the Implementation of Pharmacogenomic (PGx) Testing in Adult Mental Health Settings: A Theory-Based Systematic Review

**DOI:** 10.3390/jpm14101032

**Published:** 2024-09-27

**Authors:** Adam Jameson, Justine Tomlinson, Kristina Medlinskiene, Dane Howard, Imran Saeed, Jaspreet Sohal, Caroline Dalton, Gurdeep S. Sagoo, Alastair Cardno, Greg C. Bristow, Beth Fylan, Samantha L. McLean

**Affiliations:** 1Bradford District Care NHS Foundation Trust, Bradford BD18 3LD, UK; 2School of Pharmacy & Medical Sciences, University of Bradford, Bradford BD7 1DP, UK; 3Wolfson Centre for Applied Health Research, Bradford BD9 6RJ, UK; 4Pharmacy Department, Hull University Teaching Hospitals NHS Trust, Hull HU3 2JZ, UK; 5Leeds Teaching Hospitals NHS Foundation Trust, Leeds LS9 7TF, UK; 6Biomolecular Sciences Research Centre, Sheffield Hallam University, Sheffield S1 1WB, UK; 7Population Health Sciences Institute, Newcastle University, Newcastle NE2 4HH, UK; 8Leeds Institute of Health Sciences, Faculty of Medicine and Health, University of Leeds, Leeds LS2 9LH, UK; 9NIHR Yorkshire & Humber Patient Safety Research Collaboration, Bradford BD9 6RJ, UK

**Keywords:** pharmacogenomics, pharmacogenetics, PGx testing, precision medicine, personalised medicine, personalised prescribing, systematic review, implementation science, psychiatry, mental health

## Abstract

Pharmacogenomic (PGx) testing can help personalise psychiatric prescribing and improve on the currently adopted trial-and-error prescribing approach. However, widespread implementation is yet to occur. Understanding factors influencing implementation is pertinent to the psychiatric PGx field. Normalisation Process Theory (NPT) seeks to understand the work involved during intervention implementation and is used by this review (PROSPERO: CRD42023399926) to explore factors influencing PGx implementation in psychiatry. Four databases were systematically searched for relevant records and assessed for eligibility following PRISMA guidance. The QuADS tool was applied during quality assessment of included records. Using an abductive approach to codebook thematic analysis, barrier and facilitator themes were developed using NPT as a theoretical framework. Twenty-nine records were included in the data synthesis. Key barrier themes included a PGx knowledge gap, a lack of consensus in policy and guidance, and uncertainty towards the use of PGx. Facilitator themes included an interest in PGx use as a new and improved approach to prescribing, a desire for a multidisciplinary approach to PGx implementation, and the importance of fostering a climate for PGx implementation. Using NPT, this novel review systematically summarises the literature in the psychiatric PGx implementation field. The findings highlight a need to develop national policies on using PGx, and an education and training workforce plan for mental health professionals. By understanding factors influencing implementation, the findings help to address the psychiatric PGx implementation gap. This helps move clinical practice closer towards a personalised psychotropic prescribing approach and associated improvements in patient outcomes. Future policy and research should focus on the appraisal of PGx implementation in psychiatry and the role of pharmacists in PGx service design, implementation, and delivery.

## 1. Introduction

Psychiatric disorders are one of the leading causes of disease burden worldwide and as many as one-in-four people will experience one during their lifetime [1,2]. Pharmacological interventions help manage psychiatric disorder symptoms and prevent deteriorations in mental state. Antidepressant and antipsychotic prescribing have increased in the last two decades [3,4]. Psychotropic prescribing can be challenging, due to the vast array of options, and small differences in efficacy between them [5,6,7]. Common issues with psychotropic medicines include high rates of adverse drug reactions (ADRs) and a lack of therapeutic response [6,8,9]. Up to half of those prescribed an antidepressant will not initially respond [10], and up to a third of people prescribed antipsychotics are deemed treatment-resistant [11]. A trial-and-error approach to prescribing such medicines is, therefore, common practice, leaving patients frustrated and disheartened [12].

Personalised prescribing can be achieved by considering an individual’s genetics [13]. Pharmacogenomics (PGx) is often used interchangeably with pharmacogenetics, the former considers the simultaneous impact of multiple gene variants on medication response, while the latter relates to the impact of individual gene variants [14]. The use of ‘PGx’ in this paper refers to any gene variation impacting drug response. Genetic factors can influence drug pharmacokinetics, how the body absorbs, distributes, metabolises, and eliminates a drug [13]. Additionally, PGx can impact drug pharmacodynamics, which refers to how a drug exerts its therapeutic effect on the body [13]. Over 95% of people possess at least one gene variant that may alter their response to a given medication [15]. Those relating to the cytochrome P450 system are most relevant in psychiatry [16]. The Food and Drug Administration (FDA) publishes a list of drugs with genetic associations (drug–gene pairs), 15% of which are for psychotropic medicines [17]. The Clinical Pharmacogenetics Implementation Consortium (CPIC) and PharmGKB also list drug–gene pairs for psychotropic medicines [18,19]. PGx testing (PGxT) explores gene variants that may influence drug response, to enable PGx-informed prescribing by tailoring drug and dose selection to individuals, considering their PGx profile among other known factors in drug response [20].

Genetics explains some variability in psychotropic drug response between individuals and is thought to contribute up to 40% of response to antidepressants [21]. Several gene variants, in particular relating to HLA-A, HLA-B, and CYP2D6/2C19 genes, are associated with antidepressant, antipsychotic, mood stabiliser, and ADHD treatment outcomes [22,23,24]. Emerging evidence suggests that PGx-informed psychotropic prescribing improves the likelihood of therapeutic response and limits the risk of ADRs [25,26]. Recent meta-analyses demonstrate PGx-informed antidepressant prescribing has improved patient outcomes, increasing the chance of obtaining depressive symptom remission [27,28,29,30]. Another meta-analysis [31] revealed that PGx-informed antipsychotic prescribing may have benefits for symptom response and side effects, supporting previous research findings [32,33,34,35]. PGx-informed psychotropic prescribing may yield cost savings, owing to reduced hospitalisations, outpatient visits and prescription costs [36,37,38,39,40]. PGxT, therefore, offers an alternative to the trial-and-error prescribing approach, with potential additional health and cost benefits.

Yet to date, the implementation of PGxT in mental health (MH) settings has been limited [20]. Few countries have successfully embedded PGx to support psychotropic prescribing [41,42,43,44], leaving many that are yet to realise the potential of PGx. In the United Kingdom (UK), PGx implementation in the National Health Service (NHS) is being supported by the Genomic Medicine Service [45], but is limited to only a few clinical specialties, including dihydropyrmidine dehydrogenase testing (DYPD) for fluoropyrimidine toxicity in oncology and genotyping to prevent aminoglycoside-induced hearing loss in neonatal care [46,47].

The lack of PGx-informed prescribing in MH settings raises concern about why its utilisation is not widespread. Implementation science helps understand the gap between evidenced-based findings and their uptake in clinical practice [48]. Normalisation Process Theory (NPT) is a theoretical framework aiming to understand the ‘work’ people ‘do’ to normalise a new intervention into routine care [49]. Genomics research has traditionally lacked use of implementation theory, despite evidence that using implementation frameworks improves the translation of research findings into practice [50]. An implementation gap exists in psychiatry PGx, as widespread translation of PGx research findings into clinical practice is yet to occur. NPT offers a framework to help understand factors affecting this translational gap, and guide future research and implementation to consider the factors associated with this [51].

The aim of this review was to systematically explore the factors influencing the implementation of PGx in adult MH settings, using NPT as an underpinning theoretical framework to:Identify barriers hindering the uptake of PGx;Determine facilitators helping the adoption of PGx prescribing practices;Map key barriers and facilitators to NPT constructs to help inform future implementation of PGxT in MH settings.

## 2. Materials and Methods

The ‘Preferred Reporting Items for Systematic Reviews and Meta-Analyses (PRISMA) 2020′ [52] was used to report on this review (Appendix B) and the protocol was registered in PROSPERO on 24/02/23 (CRD42023399926).

### 2.1. Conceptual Framework

NPT includes four constructs (see Table 1), and sixteen sub-constructs that help to understand the factors influencing the success of intervention implementation [51]. It seeks to understand the ‘work’ stakeholders ‘do’ when implementing a new intervention. NPT is increasingly being used for healthcare interventions [53].

The literature on PGx implementation in practice is heterogeneous, adopting both qualitative and quantitative methods. Using NPT to underpin the research question allowed the factors influencing PGx implementation in MH to be summarised into a single, comprehensive review.

### 2.2. Search Strategy

#### 2.2.1. Phase One—Developing the Search Strategy

Using terms in a previously conducted review [54], and the ‘PICOS’ framework (see Table 2), an initial search strategy was developed and tested in EmBase. The search included three terms, ‘pharmacogenomics’, ‘mental health’, and ‘perspectives’, and synonyms for these terms, plus MeSH terms and truncations. The search strategy was further developed in consultation with two subject librarians from the University of Bradford, UK. The full search strategy can be found in Appendix C.

#### 2.2.2. Phase Two—Database Searching

The search strategy was adapted to meet the requirements of individual databases. Four databases, EmBase, MEDLINE, PsycInfo, and Scopus, were searched on 1/3/23. The research team agreed that defining a single year where PGx implementation research began would be difficult, so the review included all potentially relevant records from inception to 1/3/23. A repeat search was completed on 11/7/23, to include any relevant records published since the original search date.

#### 2.2.3. Phase Three—Reference Searching and Grey Literature

To find potentially relevant literature not identifiable via database searching, a grey literature search was conducted using Google advanced searching (see Appendix C). The reference lists of key studies from a previous review [54], were screened to identify relevant articles not identified through database searching.

### 2.3. Selection Criteria

Screening occurred in three sequential phases: (1) title, (2) abstract, and (3) full-text screening. Records were assessed using eligibility criteria (see Table 3), and if an exclusion reason could not be determined, the record entered the subsequent stage of screening to elicit more information. Rayyan, a systematic review screening software tool, facilitated screening [55]. Four independent reviewers (AJ, JT, KM, and DH) conducted screening. At each screening phase, a fifth reviewer (IS) independently assessed a 20% sample of the records screened by each of the four reviewers, to ensure consistency in eligibility criteria application. Discrepancies between reviewers following the full-text screening stage were discussed with the wider research team to reach consensus about inclusion.

A model of PGx was not specified for inclusion and instead inclusion was flexible to allow for different PGx approaches, e.g., multigene testing versus single-gene testing. We included studies at different PGx implementation stages. As psychotropic prescribing occurs in specialist and non-specialist settings, we specified that studies had to relate to PGx use in the context of psychotropic prescribing, rather than a specific setting. We included studies exploring data from different stakeholders, including healthcare professionals and patients. Finally, studies specific to PGx implementation in Child and Adolescent Mental Health (CAMH) settings were excluded because the authors agreed that influencing factors in this population are unique, warranting separate investigation.

### 2.4. Data Extraction

Microsoft Excel was used to develop a standardised data extraction tool, adapted from a tool used previously [54]. Data on barriers to and facilitators of PGx implementation, and study strengths and limitations were collected. Other data collected included the study name, author, year of publication, record type, study design, country, clinical setting, participant type, sample size, eligibility criteria, recruitment or sampling process, study aims and objectives, and study findings.

The lead reviewer (AJ) explained how to use the data extraction tool to the other reviewers (JT, KM, DH, IS). AJ completed the data extraction for each included study. Each study then had data extracted independently by either JT, KM, DH, or IS. AJ compared both datasets for each study to check for any discrepancies in study barriers, facilitators, strengths, or weaknesses. Any discrepancies were discussed between AJ and the relevant reviewer to reach a consensus.

### 2.5. Data Analysis/Synthesis

Methods of applying the NPT coding framework to assist data analysis were based on approaches taken in previous systematic reviews [56,57]. Descriptions of NPT sub-constructs were adapted, to better reflect the context of the review research question (see Table 1). Using NPT in this way provided theoretical underpinnings for each barrier/facilitator, allowing for new insights into the factors influencing aspects of PGx implementation.

Taking an abductive approach, NPT was used to tabulate extracted barriers/facilitators, and then using a codebook thematic analysis approach, adapted from the framework method [58,59], NPT assisted in theme construction. Initially, barrier and facilitator themes were mapped onto sub-constructs of NPT, before broader theme construction occurred that spanned across NPT constructs. The process of codebook thematic analysis and application of NPT occurred in five stages:Mind mapping raw barriers/facilitators to enable data familiarization;Tabulation of raw barriers/facilitators to sub-constructs of the NPT coding framework;Summarising of barriers/facilitators within sub-constructs of NPT;Theme construction of barriers/facilitators within NPT sub-constructs;Broad theme construction across NPT constructs.

The final step involving broader theme development was designed to synthesise the data into key themes that overlapped NPT constructs. See Appendix A for a summary of the approach to data synthesis using codebook thematic analysis and NPT.

During theme construction, AJ and BF discussed how different barriers/facilitators contributed to potential themes, and how barrier and facilitator themes can be summarised into main overarching themes. This was discussed further with the wider research team before the results were updated.

### 2.6. Quality Appraisal

Due to the inclusion of methodologically distinct studies, the Quality Assessment for Diverse Studies (QuADS) tool [60] was used. The tool helps to assess methodological rigour, and whether study findings are reliable, to help reach a judgement on whether the review findings are trustworthy. It includes thirteen criteria, scored between 0 and 3 to create a total score out of 39. The assessment criteria explore the extent to which studies provided a theoretical background and describe their aims, setting, and target population, and use an appropriate study design, sampling, and data collection method. It also assesses to what extent data collection and recruitment is explained, and whether the adopted data collection tool and analytic methods are justified. Finally, it assesses whether the analysis is suitable, if stakeholders were considered during design, and whether study strengths/limitations were discussed.

AJ applied the QuADS tool to each study and a second reviewer (either JT, KM, DH or IS) generated a second score, before both reviewer scores for each study were compared. When compared scores differed by <3 points, the scores were averaged and reported in the review (see Appendix A). When a score differed by >3 points, a third reviewer was involved until a consensus could be reached. No studies were excluded based on study quality, as it was useful to include barriers and facilitators from low-quality studies and identify whether these findings were corroborated by other included studies. Low-quality studies and the impact of their extracted barriers/facilitators on the review findings are discussed.

### 2.7. Patient Public Involvement and Engagement (PPIE)

A group with current MH service users and carers was consulted during the review design. They commented on the findings to inform results interpretation during data synthesis and offer insights from the service user/carer perspective during revisions of the results.

## 3. Results

### 3.1. Search Results and Included Studies

A total of 29,733 records were retrieved from four electronic databases. Following the removal of duplicates using the de-duplication tool in EndNote [61], 18,990 records entered screening, reducing to 2476 after the title screening, further decreasing to 148 records following the abstract screening. In total 29 records were selected for inclusion in the final analysis (Figure 1).

### 3.2. Study Characteristics

A summary of included study characteristics can be found in Table 4. A timeline of the included studies with key characteristics can be found in Figure 2. A range of methodologies were used, including quantitative (n = 17), qualitative (n = 9), and mixed methods (n = 3), and the studies spanned 25 countries across the continents including North America (n = 18), Europe (n = 4), South America (n = 2), Middle East and North Africa Region (n = 2), Asia (n = 2), and Australasia (n = 1). A variety of study designs were used, including surveys (n = 13), interviews (n = 6), questionnaires (n = 4), focus groups (n = 3), the Delphi method (n = 1), best–worst scaling (n = 1), and choice-format conjoint analysis (n = 1). A range of stakeholder views was explored, including healthcare professionals (HCPs) (n = 20), patients (n = 7), and both HCPs and patients (n = 2). Within the HCPs studies, the perspectives of psychiatrists-only (n = 10), mixed professionals (including psychiatrists, doctors, nurses, and pharmacists) (n = 9) and pharmacists-only (n = 1) were explored.

### 3.3. Quality Appraisal

The QuADS tool was applied to all studies, and a full breakdown of assigned scores can be found in Appendix A. The average QuADS score was 21.6 out of 39, ranging from a low-quality score of 9 to a high of 30.5. In general, some QuADS tool criteria scored higher than others across included studies.

Generally, studies described their research aims, study setting and target population in good detail, and tended to adequately describe their data collection procedures and use an appropriate method of analysis to answer their research aims. Overall, the study designs chosen, and the design of data collection tools were appropriate for addressing the research aims. Strengths and limitations were often discussed. Only one study did not provide recruitment data [64]. Explanations of the theoretical underpinning, rationale for selected data collection tools, and justification of the selected analytical method were lacking in some papers [67,69,70,74,81,85,87,89]. Appropriate sampling methods were not always applied [69,70,74,78,84,89]. Finally, some studies were particularly poor at reporting whether they had considered stakeholders during research design [62,64,65,66,67,70,74,77,78,79,81,83,85,87,89,90].

### 3.4. Barriers and Facilitators

Three key barrier themes were developed (Figure 3). A PGx knowledge gap exists among both HCPs and patients, partly due to a lack of basic and specialised training for HCPs, and a lack of awareness among patients. There is a lack of consensus in national policies and guidance on using PGx in psychiatry. Finally, there is uncertainty about the use of PGx in clinical practice, partly due to concerns about ethics, cost, and the impact of PGx.

Three overarching facilitator themes were also constructed. There is interest in the use of PGx, as a new and improved approach to prescribing. Additionally, there is a belief that PGx should be multidisciplinary with the role of pharmacy highlighted. Finally, creating the right climate for implementation helps, including integrating PGx within electronic health records (EHR), adopting local PGx champions, and engaging with staff.

The barrier and facilitator themes were mapped to constructs and sub-constructs of the NPT framework (Table 5) and are discussed in detail below.

### 3.5. Coherence

#### 3.5.1. Barriers

There was a lack of consensus and understanding about the use of PGx, its purpose, and potential benefits [65,66,67,68,75,81,87]. HCPs were unsure about when to use PGx and what PGx could achieve [68,72,75,81,90], and had fears it may replace clinical judgement when prescribing [62,67,68] or increase health inequalities [65,68]. Patients perceived PGx to be an extension of the medical model of psychiatry [64], misunderstood how PGx data are used and had unrealistic expectations of PGx [64,70,83]. There was also a belief that patients’ understanding of PGx may fluctuate [64].

Both HCPs and patients believe that PGx currently lacks clinical evidence for improving efficacy [62,65,72,79], ADRs [65,72], and medication adherence [83], and held a belief that PGx is not cost-effective [68,76,81]. They also had concerns that PGx could cause harm and may even add to existing issues in psychiatry [63,65,66,67,68,71,72,75,77,83]. Both HCP and patient stakeholders felt there was a general lack of patient awareness about PGx [68,70,73,74,75,82,84].

#### 3.5.2. Facilitators

Patients and HCPs view PGx as a valuable prescribing tool and an improvement over trial-and-error approaches [62,63,64,67,68,72,75,76,77,79,81,83,84,88,89,90], which may enhance evidence-based medicine [81]. They perceive potential benefits of PGx in depression and schizophrenia treatment following ADRs or inefficacy [65,66,67,68,69,72,75,76]. HCPs anticipate PGx helping in medication selection [63,68,81,85], dosing [65,68,71], reviews and decision-making [69,85], and help to foster patient rapport [66,67,69]. Perceived possible patient benefits include fewer ADRs [63,64,81,83,87], improved treatment efficacy [72,81,88] and quicker identification of appropriate (tolerable and effective) medication [75,76,83]. HCPs and patients perceived system benefits including reduced healthcare costs [81,83] and fewer hospitalisations [68,70]. A common belief among HCPs is that PGx may promote patient engagement during shared decision making [67,68,69,85] and provide reassurance to patients about taking medication [67,68,69,73,83,84]. Patients and HCPs agreed that PGx can help validate previous medication response [67,68,76,83,87,88].

HCPs found the experience of PGx helped them to build an understanding of what PGx requires [65,68]. HCPs also believed that not acting on or using PGx information, would not result in liability issues [65]. There was a perception among patients that additional information and counselling received as part of PGx helped them feel more informed about their medication and illness [68,77,84].

### 3.6. Cognitive Participation

#### 3.6.1. Barriers

Patients and HCPs both expressed concerns about PGx, highlighting ethical issues such as obtaining consent, genetic data storage, and confidentiality [62,63,64,68,70,72,75,83,84]. They both anticipated challenges in integrating PGx into existing clinical pathways [69,70,88]. Some psychiatrists felt unequipped to lead implementation due to a lack of skills, knowledge, and confidence in PGx [68,75,78,87]. Certain sub-populations of psychiatrists, including older or more experienced psychiatrists [78] or those with a psychosocial approach [75], were less inclined to consider PGx. Some psychiatrists perceived their colleagues to lack the necessary knowledge to use PGx [63]. Furthermore, some psychiatrists were worried about counselling patients about PGx [75,88] and updating PGx results based on evolving evidence [69,83].

#### 3.6.2. Facilitators

HCPs and patients had a strong interest in PGx due to a range of perceived potential benefits [62,65,66,72,73]. Education is believed to be important, to improve understanding of PGx [65,68,70,76,80,82,86]. HCPs agreed on the tasks required to initiate and sustain PGx [68,69,71,72] and believed that EHR can facilitate these activities [67,68,86]. HCPs were willing to collaborate with others on PGx [68,86,90], and trust was perceived to be key to this [69,90]. Pharmacists were highlighted to be important during the implementation [62,68,90], and pharmacists viewed PGx as part of their expanding clinical role and felt equipped to contribute to PGx activities [90]. Genetic counsellors were perceived to have an important role in discussing PGx results with patients [84,85]. Psychiatrists saw PGx within their scope of practice, particularly in offering PGx to suitable patients [66,71,72]. A multidisciplinary team (MDT) approach to monitoring outcomes from PGx was desired [86,90].

### 3.7. Collective Action

#### 3.7.1. Barriers

Both HCPs and patients expressed scepticism about PGx, with concerns about the cost [62,63,66,67,68,72,75,76,81,83,87,88] and the possibility of misinterpreting results [68,76,88]. Both stakeholders noted that PGx could hinder the development of patient–clinician rapport [62,63,75]. HCPs were apprehensive about PGx’s ability to enhance outcomes [65,68,85], especially in influencing psychiatrists’ prescribing decisions [75,85]. HCPs cited concerns including the additional time required for PGx tasks [67,68,69,72], potential delays in obtaining results [67,68,70,74,75,83,88], lengthy PGx reports [68], and the need for extra support to use PGx results [63,67]. A knowledge gap among HCPs was evident [66,68,75,81,87], with psychiatrists lacking awareness of PGx guidance [62,63,65,66,75,87,88] and holding a perception they are inadequately educated to use PGx [66]. Patients also perceived HCPs as lacking knowledge and awareness of PGx [70,73,74,82,83], and patients were also thought to have insufficient education and awareness about PGx [70,73,74,82]. HCPs additionally highlighted a perceived lack of policy [71], basic and specialised training [62,63,68,72,75,90], and guidance and resources for implementing PGx [62,63,65,71,87].

#### 3.7.2. Facilitators

HCPs perceived PGx to be a promising new strategy that will change prescribing and become a standard practice [62,66,71,72,75,87,89,90], which can help make prescribing easier when delivered in a timely, accessible manner [74]. A belief that PGx is safe and reliable [62,68,72,75], and relevant training and experience equipped HCPs with the skills and knowledge to develop confidence to use PGx [65,66,72,80]. HCPs believe that psychiatry-specific, practical guidelines [65], and a national PGx policy would help implementation [63,65,86]. Staff and leadership engagement and adoption of local PGx champions facilitated the implementation [80,86]. Some pharmacists perceived themselves as competent in identifying clinical situations to offer PGx and in counselling patients on PGx [66,90]. Both HCPs and patients were enthusiastic to learn more about PGx [62,66,70,75] and believed that education informs patients and manages their expectations. PGx is perceived to help foster clinician–patient rapport [66,67,83].

### 3.8. Reflexive Monitoring

There was a lack of facilitator and barrier themes linked to the reflexive monitoring construct of NPT. However, HCPs reported using their own clinical judgement to appraise outcomes from implementing PGx, and there was a preference to adapt PGx for specific contexts based on experience and feedback [88].

## 4. Discussion

### 4.1. Key Findings

This novel systematic review explores barriers and facilitators to implementing PGxT in MH settings. By using NPT as an underpinning theoretical framework, our findings offer new insights, revealing barrier and facilitator themes constructed within and distributed equally across the coherence, cognitive participation, and collective action NPT constructs. Thus, outlining that NPT can help understand factors influencing PGx implementation. However, there was an absence of barriers and facilitators within the reflexive monitoring construct. This review answers a call for the exploration of how PGx can be implemented in a context-specific manner, considering the needs of distinct patient populations and clinical specialties [91]. The review also highlights the potential for using NPT as a theoretical framework to explore and understand PGx implementation in MH.

Notably, our findings expose a potential oversight in the consideration of methods, processes, or tools evaluating PGx implementation. No barriers within the reflexive monitoring construct were established, a finding also identified by an NPT-informed systematic review exploring deprescribing interventions [57]. We speculate this may stem from limited exploration of reflexive monitoring within the research literature, rather than an absence of barriers. Reflexive monitoring facilitators included the belief that HCPs use their clinical judgement to appraise outcomes following PGx and wish to adapt PGx use based on experience. We propose that a standardised process for evaluating PGx implementation has yet to be developed, despite the need for evaluating complex intervention implementation [92]. Several concerns about PGx in MH were noted, but without an effective process for monitoring outcomes following implementation, it is difficult to demonstrate how concerns have been addressed or identify new emerging concerns.

The findings highlight that a lack of education and training (E&T) opportunities for HCPs has contributed to a knowledge gap. This reflects the insufficient knowledge of PGx across healthcare professions [93,94]. E&T is often a challenge when implementing new healthcare interventions [95,96]. This barrier can be addressed nationally and locally through increased PGx teaching during undergraduate training [97,98,99], a strategy for educating qualified HCPs. This is relevant to MH, to upskill psychiatrists, nurses, and pharmacists to communicate PGx to patients and use PGxT during medicines optimisation [100]. The NHS Genomic Medicine Service is actively addressing this challenge in the UK [45]. Opportunities to raise patient awareness should be sought, by developing tailored resources to enhance understanding of PGx application in MH. The included study by Sloat et al. [82] demonstrated this is possible, by creating a pre-PGx test educational video that improved knowledge of, attitudes towards, and perceived control over PGx.

Uncertainty towards PGx was established, particularly regarding ethical concerns, PGx cost, and its potential impact. Funding to commission PGx service development, with equitable access independent of socioeconomic status, is necessary [101]. Finances can prevent the widespread uptake of new evidence-based findings [102]. A review of psychiatric PGx economic evaluations reported favorably on PGx being cost-effective or cost-saving compared to standard care [37]. However, the included evaluations were heterogeneous, and the authors called for a more extensive evaluation of the health economics in psychiatric PGx. Regarding the cost-effectiveness of broader PGx implementation, a recent review found that once implemented the majority of CPIC guidelines were cost-effective [103], supporting previous findings that PGx was likely to be cost-effective, especially if the cost of genotyping continues to reduce [104]. More dynamic models for determining cost-effectiveness are required [105], especially given the potential for next-generation sequencing approaches in the future [106,107]. In the UK, the current evidence focuses on single or small numbers of genes associated with one drug [108,109]. This potentially undervalues PGx, as PGx results may be applicable beyond psychiatry during prescribing in other clinical areas. Linked to cost, are the finances associated with developing laboratory infrastructure necessary to implement PGx. The UK’s PROGRESS project is assessing the rollout of PGx to GP practices and will provide insights into how to build infrastructure to deliver PGx at scale [110].

The turnaround time of PGx results was noted as a barrier. The impact of PGx on clinical practice may differ, depending on how quickly results are available. Engaging with stakeholders to understand acceptability towards the length of time for PGx results turnaround will be important. Ethical concerns were also noted and are understandable given the sensitivity toward using genetic data [111]. Previous research identified confidentiality concerns associated with using genomics in healthcare [112]. Robust privacy and data provisions for storing and using PGx data, along with effective education about how PGx data are used, will be necessary to alleviate these concerns. PGx training can also address uncertainty about the efficacy and cost-effectiveness of PGx, by introducing relevant evidence to HCPs [36,37,103,113,114]. Through engagement with patient stakeholders, materials to adequately address concerns about PGx can be co-designed [115].

Facilitators included the belief that PGx is a new, improved prescribing approach and highlighted the importance of creating an implementation climate—by engaging staff, identifying local PGx champions and enabling full EHR integration. Previous implementation research showed that local champions can positively influence new intervention uptake [102]. Therefore, organisations should empower local champions, who will be specialists in psychiatry, with additional expertise on the impact of PGx in MH. Pharmacists are ideally placed professionals, due to their involvement in prescribing and medicine pathways. PGx champions can act as knowledge brokers [116], ensuring organisations are up to date with PGx developments, engaging staff when developing PGx services, and providing necessary PGx training for the wider workforce. Organisations should aim to integrate PGx information within EHRs and clinical decision support systems. Findings from the PROGRESS trial [117] will be key to initiating this within the UK NHS and will serve as a template for how PGx can be integrated into a broad, complex health system.

There were similarities and differences in the perceived barriers and facilitators between HCPs and patients. In the coherence construct, there was agreement on how both stakeholder groups made sense of PGx and distinguished PGx from routine practice. HCPs emphasised that experience using PGx enhances understanding, and expressed few liability concerns if PGx results are not used. In the cognitive participation construct, HCPs held more views on who to offer PGx to, while both stakeholder groups highlighted challenges in integrating PGx into practice, citing ethical concerns and consent issues. Likely due to their job roles, HCPs expressed more opinions on PGx-related tasks and responsibilities compared to patients. Within the collective action NPT construct, both stakeholder groups mentioned the importance of integrating PGx into routine patient conversations, but HCPs focused more on ensuring staff are well-equipped with the appropriate skills and knowledge to implement PGx.

### 4.2. Future Policy and Research

PGx has the potential to revolutionise psychotropic prescribing, disrupting established prescribing processes and traditional prescriber–patient dynamics [20]. This shift in the status quo will not occur without challenges [118], and there are several barriers to implementing PGx in MH. These challenges can be better understood and addressed if evidence-based implementation strategies are used to facilitate the uptake of emerging evidence about the benefits of PGx-informed psychiatric prescribing [119]. We demonstrated that NPT can be used to explore PGx integration into routine care. Using NPT in future PGx research will provide further insights about what contributes to successful implementation, at both the system and individual level. We suggest using NPT in future studies exploring PGx clinical utility and future research that qualitatively assesses contextual factors influencing implementation. This may help address why PGx research findings are not being widely embedded into practice and enable policymakers, service providers, and HCPs to successfully develop PGx services.

The use of NPT identified a lack of factors within the reflexive monitoring construct and the field would benefit from building an understanding of factors within this construct. This could be achieved by developing systems to appraise PGx implementation, allowing a universal approach to evaluation. Tools collecting data post-implementation, and exploring stakeholder and system outcomes, such as impact on prescribing, beneficiaries of PGx, experiences and general satisfaction with PGx, would facilitate evaluation. The field may also benefit from a process evaluation [120], to explore contextual factors influencing how PGx interventions are delivered in MH. This would deepen understanding of implementation and determine where improvements are required. NPT can support process evaluation, by enhancing understanding of processes that influence how PGx is integrated in practice. Such reflection will be critical to the long-term success of PGx.

Further investigation is warranted to explore why ethical and consenting concerns appear to be more specific to psychiatry, than PGx implementation in primary care [91]. Guidance on obtaining consent to conduct PGx and avoid genetic discrimination should also be considered and developed. A lack of consensus within national psychiatry guidelines was a barrier to PGx implementation. In the UK, no guidance exists and the Royal College of Psychiatrists (RCPsych) [121] recently recommended against using PGx. Despite citing no supporting evidence, they did report a lack of research regarding how and where to use PGx. This research gap can be addressed using implementation frameworks, to bridge the void between emerging clinical evidence supporting psychiatric PGx and an understanding of how to implement PGxT in different MH settings. We propose using NPT for such research, to explore how to implement PGx in different MH settings. A UK-specific PGx implementation policy and guidance, developed by organisations such as the RCPsych and the British Association for Psychopharmacology (BAP), would help address local concerns and adapt CPIC guidelines for regional use. Some countries have performed this already, demonstrating that it can be achieved [122,123,124].

Like previous findings, the potential role for pharmacists in PGx was identified, as they are anticipated to play a key role in PGx implementation [99,125]. We found that HCPs are keen to collaborate with pharmacists and that pharmacists are enthusiastic about delivering PGx in psychiatry. Further exploration of pharmacy-led PGx implementation models is necessary, building on the work by Weinstein et al. [90]. A barrier in this review was the psychiatrist’s concern over the potential burden of PGx. In the UK, where pharmacists are increasingly becoming prescribers and will soon enter the workforce as qualified independent prescribers [126], a pharmacist-led PGx model could expand their clinical role and address psychiatrist concerns. A potential role for genetic counsellors in PGx was also identified, prompting further research into genetic counsellors roles in the broader uptake of genomics, not specific to pharmacogenomics [127]. A summary of policy, practice, and research recommendations can be found in Table 6. 

### 4.3. Strengths and Limitations

The findings are highly applicable to PGx in depression, as many included studies focused on PGx-informed antidepressant prescribing. The findings are potentially less applicable to other psychotropic classes, including antipsychotics, ADHD medicines and mood stabilisers, despite all having known pharmacogenomic associations [22,23,24]. This may reflect the variability in evidence for PGx clinical utility across different psychiatric disorders [28,29,30,31]. That said, in naturalistic study settings, patients often have psychiatric co-morbidities, and it is common for these populations to be concomitantly prescribed different psychotropic drugs. Real-world use of PGx is more complex than PGx clinical trials, which often use single or small gene panel PGxT, in a single psychiatric disorder. This potentially explains why panel-based PGx approaches appear to be preferred in MH settings [24]. Nonetheless, exploring PGx in more diverse conditions is necessary, to understand the perspectives of people with psychosis, schizophrenia, bipolar disorder, and ADHD. Gene testing in psychiatry does have general limitations too. Genetics are just one factor that affects diagnosis and drug response in the bio-psycho-social model of care [128]. Some gene tests lack clinical validity [129,130], and as advancements in genomic sequencing enable increased associations between genes and psychiatric disorders, numerous confounding factors in clinical practice impact these findings [131,132].

The quality of included studies varied, and caution should be exercised when drawing conclusions from barrier and facilitator themes developed from a single or small number of included studies. A lack of confidence in PGx effectiveness and older/more experienced psychiatrists being less likely to consider PGx, were both identified barriers by McMichael et al. [78]. This was a low-quality study due to its poor design, inappropriate sampling, and lack of detail in the study aims, setting, and recruitment. Similarly, Kung et al. [74] was deemed low quality, because the study description lacked depth, likely because it was a ‘letter to the editor’, rather than an original research article. Barriers and facilitators extracted from this paper were corroborated by other studies, except for the facilitator that PGx makes prescribing easier when delivered in a timely manner, which should be interpreted with caution. Walden et al. [89] also achieved poor QuADS tool scores but extracted barriers and facilitators were corroborated by several other studies.

Another drawback was in studies where PGx had been implemented, due to a risk of bias, as authors may have wanted to validate their effort implementing PGx by reporting beneficial outcomes. There was high heterogeneity among the included studies, as they were conducted in a range of settings with diverse participants. This makes interpretation of the review findings difficult and suggests that the review findings are not necessarily applicable to all MH settings that implement PGx. The review findings are limited by the model of PGx implementation used or proposed within included studies and are not necessarily generalisable to all PGx models. Alternate models are possible, by using different PGx tests (single-gene versus panel-gene tests) and personnel during implementation. In some countries, pharmacists are not able to prescribe, and their clinical role is minimal. Therefore, some findings relating to the input of pharmacists during PGx implementation are not relevant to all countries. Highlighted limitations indicate that higher quality research is required to inform the future of psychiatric PGx implementation.

This review is the first to systematically summarise the barriers and facilitators to implementing PGx in MH settings. It ties together a body of literature about the perspectives, opinions, and views of HCPs and patient stakeholders toward the use of PGx in psychiatry. A further strength is the use of the NPT as an underpinning theoretical framework that has helped to highlight where gaps in the literature exist that warrant further exploration. To our knowledge, this is the first time NPT has been used in PGx implementation research and we propose that it can be used in future studies to inform implementation research by examining how PGx is integrated and normalised in clinical practice.

## 5. Conclusions

Taking an abductive approach, this review used NPT to construct barrier and facilitator themes about the implementation of PGx in MH settings. We found that there is a PGx knowledge gap among stakeholders, uncertainty about the use of PGx and a lack of guidance on the use of PGx in psychiatry. We also found that there is an interest in the use of PGx, with the belief that a multidisciplinary approach and EHR integration can aid implementation. Using NPT as a theoretical framework enabled novel insights into factors influencing PGx implementation in MH by systematically synthesising the literature in the field to date. We propose that NPT should be used as an implementation framework in future PGx research, and that future research should address the highlighted gaps in NPT’s reflexive monitoring construct. Policy makers should consider key barriers, such as a lack of PGx education opportunities and a lack of guidance on the implementation of PGx, as demonstrated by the review.

## Figures and Tables

**Figure 1 jpm-14-01032-f001:**
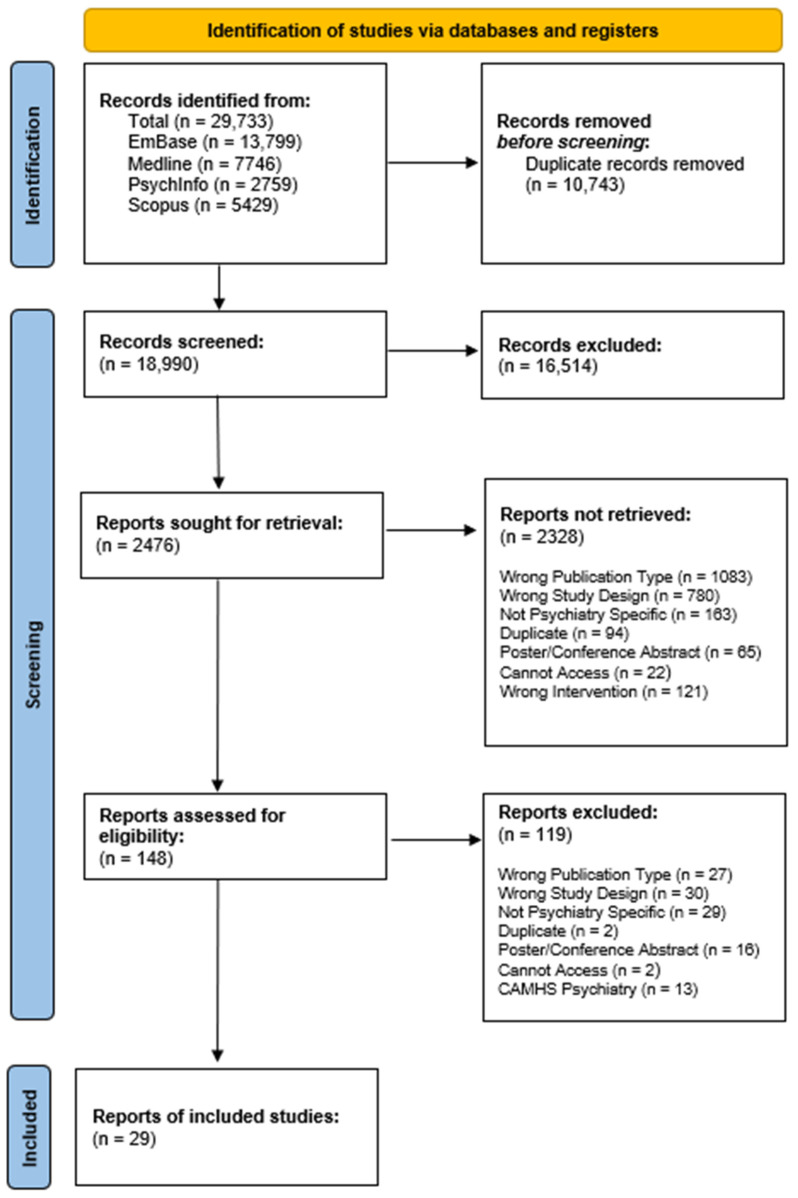
PRISMA flow chart [52]. Description: pictorial diagram of the process used for identifying and screening relevant records for inclusion, following PRISMA guidance.

**Figure 2 jpm-14-01032-f002:**
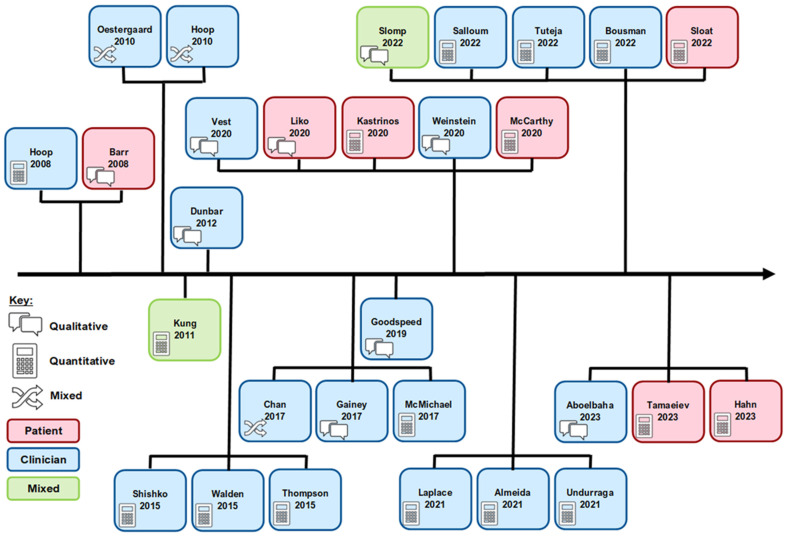
Timeline and key characteristics of included studies. Description: A timeline of the included studies showing the study lead author name, study publication year and a brief description of methods used. Those shown in blue are healthcare professional (HCP) only studies, those in red are patient-only studies and those in green are studies including both HCPs and patients. The icons represent whether the study used quantitative, qualitative, or mixed methods approaches. For information about barriers and facilitators extracted from each individual study please refer to Appendix A.

**Figure 3 jpm-14-01032-f003:**
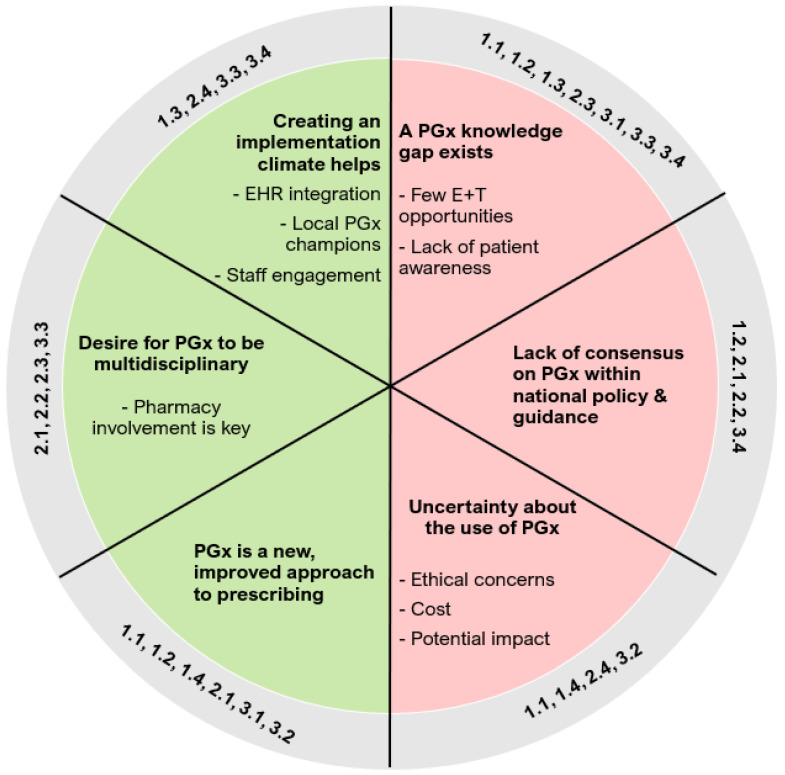
Key barrier and facilitator themes with associated NPT sub-constructs. Description: a summary of the main facilitator (left) and barrier (right) themes and associated NPT sub-construct numbers, constructed from the barriers and facilitators presented in Table 5. ‘E + T’ = education and training; ‘EHR’ = electronic health record; ‘PGx’ = pharmacogenomics.

**Table 1 jpm-14-01032-t001:** Normalisation Process Theory (NPT) framework.

Coherence	Cognitive Participation	Collection Action	Reflexive Monitoring
1.1—Differentiation: sense-making work that is carried out to understand how PGx practices and the PGx approach is different from standard prescribing	2.1—Initiation: relational work people do to drive forward the use of PGx practices	3.1—Interactional Workability: work that people do with each other, objects and factors relating PGx when operationalising in practice	4.1—Systematisation: work of stakeholders in collecting information about PGx to determine how useful it is for themselves and others
1.2—Communal Specification: sense-making work that people do together to build a shared understanding of the aims, objectives, and potential benefits of PGx	2.2—Enrolment: work involved in reorganising people, their individual and group relationships, to collectively contribute to the use of PGx	3.2—Relational Integration: knowledge work that people do to build accountability and maintain confidence in PGx and others as they use it	4.2—Communal Appraisal: the work that people do collectively to evaluate the worth of PGx in practice
1.3—Individual Specification: sense-making work people do individually to understand the tasks and responsibilities relating to PGx expected of them	2.3—Legitimation: relational work that people do to ensure that others believe they are right to be involved in PGx and that they can contribute to its use in practice	3.3—Skill Set Workability: allocation work to divide the labour of PGx and related work, as PGx is operationalised in practice, based on individual and group skills and knowledge	4.3—Individual Appraisal: work people do individually to appraise how PGx impacts on them and their work contextually, to express their personal relationship to PGx
1.4—Internalisation: sense-making work people do to understand the worth of PGx, including its value, benefits, and importance	2.4—Activation: work that people do to collectively define the actions and processes required to sustain PGx use	3.4—Contextual Integration: resource work carried out to allocate resources, and execute policies, procedures and protocols to initiate the use of PGx in practice	4.4—Reconfiguration: appraisal work people do to redefine or modify PGx practices or procedures

Description: Explanation of the sub-constructs within the NPT [49] framework, in the context of pharmacogenomics, adapted from May et al. [53].

**Table 2 jpm-14-01032-t002:** PICOS (Population, Intervention, Comparison, Outcome, Study Type) framework.

PICOS Criteria	Explanation
Population	Healthcare professionals working in mental health settings OR patients with a psychiatric disorder cared for by mental health care providers.
Intervention	The use of PGx Testing during prescribing of a psychotropic medicine.
Comparison	** Not applicable.*
Outcome	Primary qualitative or quantitative data collected pre-, post-, or during PGx implementation, in the context of psychotropic prescribing.
Study Type	Qualitative or quantitative studies

Description: Breakdown of the PICOS criteria was used to specify the search strategy deployed during the review screening process. * A comparator was not required, as the review did not aim to assess PGx clinical effectiveness, and instead explored barriers and facilitators to implementing PGx.

**Table 3 jpm-14-01032-t003:** Inclusion and exclusion criteria.

Inclusion	Exclusion
Data from qualitative and/or quantitative studies relating to healthcare professional and/or patient stakeholder factors (barriers and facilitators) influencing implementation of PGx within mental health settings (defined as any setting where psychotropics are prescribed).	Data about factors affecting implementation of PGx testing in non-mental health settings. Studies exploring PGx implementation in child and adolescent mental health (CAMH) settings. Randomised controlled trials that do not include data about barriers and facilitators to PGx implementation. Systematic Reviews/Narrative Reviews Conference abstracts or oral presentations not available in full text

Description: Eligibility criteria used during screening of records for inclusion.

**Table 4 jpm-14-01032-t004:** Summary of included studies characteristics.

Author and Year	Aims	Design, Methods, and Country	Study Setting and Participants	Funding Source and Conflicts of Interest
Aboelbaha, 2023 [62]	To explore the knowledge, level of engagement, and perspectives on the use of PGx testing when making depression management decisions among practicing psychiatrists in the MENA region	Qualitative Semi-structured interviews Saudi Arabia, Qatar, Bahrain, Jordan, Egypt, Tunisia, Oman, Palestine, Iraq, Kuwait, UAE, Algeria	Mental health settings— specifically depression management Psychiatrists	No conflicts of interest were declared by the authors. Funding was via open-access funding provided by the Qatar National Library.
Almeida, 2021 [63]	To assess the perception and knowledge of PGx among Brazilian psychiatrists	Quantitative Cross-Sectional Survey Brazil	Clinical setting not specified Psychiatrists	No conflicts of interest were declared. Lead author is a recipient of a public scholarship, but the research did not receive any specific grant funding.
Barr, 2008 [64]	To explore the range of factors that may impinge upon public and service user acceptability of the pharmacogenomics of antidepressants	Qualitative Focus Groups England, Poland, Germany, Denmark	Mental health settings— specifically depression clinics Patients General Public	The research project was funded by the European Commission. No statement on conflicts of interest stated included.
Bousman, 2022 [65]	To explore the perceptions toward PGx testing among members of the American Association of Geriatric Psychiatry	Quantitative Cross-Sectional Survey USA	Inpatient and outpatient geriatric settings Psychiatrists	Conflicts of interest declared by some authors, with some in receipt of a range of public and private funding.
Chan, 2017 [66]	To assess the attitudes and opinions of clinicians in psychiatry in Singapore towards pharmacogenomic testing, and in doing so elicit its possible barriers and risks to employ this technology in patient care	Qualitative and Quantitative Web-based survey Singapore	Public and private mental health settings Psychiatrists Pharmacists	No conflicts of interest declared by authors. Study was supported by the Singapore Ministry of Health’s National Medical Research Council.
Dunbar, 2012 [67]	To explore experiences of ordering, receiving, and utilizing the AmpliChip^®^ CYP450 test results, as well as the perceived advantages and disadvantages of employing the test in practice.	Qualitative Interviews New Zealand	Secondary care mental health settings Psychiatry doctors	No conflicts of interest statement included. The study was funded by Roche Diagnostics, through an unrestricted research grant, with 100 of the AmpliChip CYP450 tests being made available free of charge.
Gainey, 2017 [68]	To evaluate mental health clinicians’ perceived knowledge regarding pharmacogenetic testing, their attitude, receptivity towards, and confidence in pharmacogenetic testing, and how pharmacogenetic testing is being implemented to support precision medicine in outpatient clinics	Qualitative Semi-structured interviews USA	Outpatient mental health clinics Nurse Practitioners, Clinical Nurse Specialists, Physician Assistants Medical Doctors (certified in psychiatry)	Dissertation completed in fulfilment of a PhD. No conflicts of interest or funding statements provided.
Goodspeed, 2019 [69]	To evaluate input from mental health clinicians on electronic health record integrated clinical decision support (CDS) tool and PGx, and the reactions of psychiatry clinicians to a CDS prototype	Qualitative Focus Group USA	Clinical setting not specified Doctors Nurses (That had psychiatry certification)	The study was funded through a grant from the National Institute of Mental Health. Several authors are or were employees of RxRevu, a for-profit healthcare information technology company.
Hahn, 2023 [70]	To identify barriers to implementation of PGx in Germany, to identify why implementation has been slower in other countries	Quantitative Questionnaire Germany	Inpatient depression clinics Patients	
Hoop, 2008 [71]	To investigate the attitudes of a random national sample of psychiatrists about the likely impact of genetic testing on psychiatric patients and the field	Quantitative Survey USA	Mix of clinical settings—inpatient and outpatient psychiatric settings, both public and private Psychiatrists	Funding from the National Institutes of Health declared. No conflicts of interest statement included.
Hoop, 2010 [72]	To systematically assess attitudes and experiences of psychiatrists regarding the role and key clinical/ethical issues relevant to Pharmacogenetic Testing in psychiatry	Qualitative and Quantitative Survey USA	Academic medical centres with Departments of Psychiatry Psychiatrists Psychiatry Residents	No conflicts of interest reported by authors. Funding declared from the ‘Research for a Healthier Tomorrow-Program Development Fund’.
Kastrinos, 2020 [73]	To understand what psychiatry patients already know about the use of PGx in psychiatry and their interest in participating in testing.	Quantitative Questionnaire USA	Clinical setting not specified Patients	Research supported by a National Institute for Health award. No conflicts of interest statement included.
Kung, 2011 [74]	To explore patient and clinician satisfaction with Pharmacogenetic Testing	Quantitative Survey USA	Inpatient psychiatry Clinicians Patients	One author is an employee of AssureRx, a personalised medicine company, but not at the time of the study. AssureRx also provided genotyping testing as part of this study.
Laplace, 2021 [75]	To evaluate the acceptability of PGxT by psychiatrists and psychiatry residents in France using a four domains acceptability model based on International Organisation for Standardization (ISO) and Nielsen models (usefulness, usability, easiness, and risk).	Quantitative Survey France	Mix of clinical settings—both inpatient and outpatient psychiatry, including adult, geriatric, child and adolescent, substance misuse and forensic psychiatry Psychiatrists Psychiatry Residents	No external funding was received to conduct the research. Authors reported no conflicts of interest.
Liko, 2020 [76]	To assess patients’ perspectives and experiences with psychiatric pharmacogenetic testing	Qualitative Semi-structured interviews USA	Outpatient psychiatry— depression Clinic Patients	No conflicts of interest reported by authors. No statement about the funding of research.
McCarthy, 2020 [77]	To assess motivations, attitudes, and concerns about PGxT in a cohort of depressed veteran patients with past drug treatment failure indicating some degree of treatment resistance using the MAPP instrument	Quantitative Questionnaire USA	Secondary Care Psychiatry Patients	Funded through an award from the National Institutes for Health. No conflicts of interest declared by authors.
McMichael, 2017 [78]	To contribute to the topical issue of whether genotype information influences the treatment recommendations of psychiatrists when a patient’s treatment response (in terms of symptom improvement) is already known to the psychiatrist.	Quantitative Choice-format conjoint analysis (discrete choice experiment) Northern Ireland	Clinical setting not specified Psychiatrists	No conflicts of interest reported by authors. Financial support declared and was provided through a grant from the Department of Education and Learning.
Oestergaard, 2010 [79]	To provide expert perspectives regarding the extent to which the introduction of 5-HTTLTR pretesting in clinical practice as a routine procedure would lead to better clinical outcomes	Qualitative and Quantitative Delphi Method Not Specified	Clinical setting not specified Experts in 5-HTTLPR genotyping (doctors and pharmacists)	No conflicts of interest or funding statements included in the paper.
Salloum, 2022 [80]	Using a BWS experiment to evaluate the importance of implementation factors for PGx testing to guide antidepressant prescribing	Quantitative Best Worse Scaling Survey USA	Clinical setting not specified—participating organisations were funded and affiliate members of the IGNITE Network (a multidisciplinary consortium focused on the development, implementation, and dissemination of methods that integrate genomic medicine into clinical care) Individual participant’s roles not specified	Some authors reported a combination of funding and associations with private and public organisations. Research in the publication was funded through several research grants from a variety of public organisations and institutions.
Shishko, 2015 [81]	Evaluate psychiatric pharmacists use, knowledge, and perception of the effectiveness of PGx testing	Quantitative Cross-sectional survey USA, Canada, Abu Dhabi, Indonesia, Singapore	Mix of clinical settings—inpatient and outpatient settings, both public and private, and community pharmacy Psychiatric Pharmacists	No conflicting interests were reported by the authors and no industry funding was used in the research.
Sloat, 2022 [82]	To assess perspectives of patients with depression on PGxT for depression management and study the impact of an educational intervention for this population	Quantitative Case–control survey study USA	Clinical setting not specified Patients	University funding declared. Authors reported no conflicts of interest.
Slomp, 2022 [83]	To explore the perceptions if PGxT among PWLE and P/HCP to inform the process of considering the clinical implementation of PGxT for depression within the healthcare system in British Colombia (BC), Canada.	Qualitative Semi-structured interview Canada	Clinical setting not specified People with lived experience (PWLE) Professional stakeholders (clinicians, laboratory staff, insurance representatives, and policy makers)	Authors reported no conflicts of interest. Public award funding was declared.
Tamaeiev, 2023 [84]	To learn more about psychiatric patients’ attitudes towards PGx	Quantitative Survey USA	Inpatient and outpatient psychiatric Settings Patients Patient family members	Some authors reported conflicts of interest with affiliations to Genomind Inc and InformedDNA. Several authors declared public funding through the National Institutes for Health.
Thompson, 2015 [85]	To assess attitudes towards integration of genetic counselling into psychiatric patient care, specifically in the context of the use of pharmacogenomic test results to guide treatment.	Quantitative Cross-sectional questionnaire study USA	Inpatient and outpatient psychiatric settings Psychiatrists Psychiatry residents	Authors reported no conflicts of interest. Funding came through a gift donation and a grant from the National Society of Genetic Counselor’s Psychiatry Special Interest Group.
Tuteja, 2022 [86]	To understand factors important for the implementation of PGxT to guide antidepressant prescribing	Quantitative Survey USA	Clinical setting not specified—17 sites that had either implemented PGx or were planning to	Research was supported through funding from a range of public funding bodies, mainly the National Institutes for Health. Several authors reported affiliations to or funding from private organisations.
Undurraga, 2021 [87]	To explore opinions about current practices, perceived value, and barriers to clinical use of PGxT amongst Chilean psychiatrists	Quantitative Survey Chile	Mix of clinical settings—including child, adolescent, and adult psychiatry settings in both public, private, and academic sectors Psychiatrists	Funding was provided by public agencies. The research was conducted in absence of any commercial or financial conflicts of interest.
Vest, 2020 [88]	To understand providers’ perspectives of PGx for antidepressant prescribing and implications for future implementation	Qualitative Focus Groups USA	Outpatient psychiatric clinics and primary care clinics Psychiatrists Primary Care Providers (Internists, Family Medicine, Advanced Nurse Practitioners)	Myriad Genetics provided in-kind testing for the study. Funding reported from a range of public organisations. A range of affiliations to different commercial and private organisations declared by one author.
Walden, 2015 [89]	To explore physicians’ opinions of PGxT and their experiences using PGxT for psychotropic medication	Quantitative Survey Canada	Mix of clinical settings—psychiatric and primary care settings Psychiatrists General Practitioners	Author funding reported from a variety of public organisations. One author reported affiliations with private companies
Weinstein, 2020 [90]	To explore physicians’ and pharmacist stakeholder perceptions on implementing a pharmacist-run pharmacogenomic service for patients with depression in a primary care setting	Qualitative Semi-structured interviews USA	Primary care outpatient family medicine practices Pharmacists Family Medicine Physicians	Funding through a grant from the Pennsylvania Pharmacists Association Educational Foundation. No conflicts of interest reported.

Description: The table displays characteristics of included studies in the systematic review, including the study title, author, year of publication, aims, country, design and methods, clinical setting and participants, source of funding, and stated conflicts of interest. PWLE (People with Lived Experience); P (Patient); HCP (Healthcare Professional).

**Table 5 jpm-14-01032-t005:** Barriers and facilitator themes and sub-themes mapped to NPT constructs and sub-constructs.

NPT	Barriers		Facilitators	
1. **Coherence**				
1.1 Differentiation	HP	Fear PGxT may replace (rather than complement) existing prescribing practices [62,67,68]	B	Perception that PGx offers an improved approach to prescribing [62,64,88]
	Pa	Perception that PGxT is an extension of the medical model of psychiatry [64]	B	Better than trial-and-error approach [62,68,83,88,90]
			B	May improve evidence-based medicine in psychiatry [81]
1.2 Communal Specification	B	Lack of consensus about purpose and potential benefits of PGxT [65,67,75,81,87]	B	Perception that PGx is a tool to help guide prescribing and support clinical decisions to improve prescribing outcomes [62,63,67,68,72,75,76,77,79,81,83,84,89]
	HP	HCPs: when and who to offer PGxT, and disagreement about what PGxT can achieve and how information is used [68,72,75,81,90]	B	Particularly for prescribing in depression and schizophrenia following ADRs or inefficacy [65,66,67,68,69,72,75,76]
	Pa	Patients: misunderstanding of how PGx data is used (e.g., in diagnosis or prognosis) and unrealistic expected benefits for PGxT [64,70,83]	Pa	Additional information and counselling given to patients during PGxT helps feel more informed about their medication and illness [68,77,84]
1.3 Individual Specification	HP	Lack of understanding about what PGxT entails and requires of them [66,67,68,81,87]	HP	Experience helped stakeholders understand what PGx requires of them [65,68]
	B	Patient lack of awareness about what PGxT is [68,70,73,74,75,82,84]	HP	Both personal experience and access to those with experience of using PGxT [68]
	Pa	Patient understanding of PGxT involves for them may fluctuate based on mental health status [64]	HP	Prescribers believed not acting on or using PGx results would not cause liability issues [65]
1.4 Internalisation	B	Belief that PGx currently lacks evidence to support clinical utility, specifically for: [62,65,72,79,87,88]	B	Perception that PGx has a range of potential HCP, individual, and system benefits and values: [62,63,81,83,84,87,88]
	HP	Efficacy [62,65,72,79]	HP	HCP—help inform medicine choice [63,68,81,85] and dose [65,68,71], help build rapport with patients [66,67,69], inform medication reviews and clinical decisions [69,85]
	HP	Adverse Drug Reactions [65,72]	B	Patient—⇩ADRs [63,64,81,83,87], ⇧efficacy [72,81,88] and ⇩time to find suitable medicine [75,76,83]
	B	Adherence [83]	B	System—⇩health costs [81,83], and ⇩frequency [70] + ⇩length of hospital admissions [68]
	B	Concern that PGxT may cause harm or distress [63,65,66,67,68,71,72,75,77,83]	HP	Belief that PGx can help engage patients in shared decision making [67,68,69,85]
	HP	Increase health inequalities [65,68]	B	Agreement that PGx can reassure patients by reducing uncertainty about taking a medicine [67,68,69,73,83,84]
	B	Perpetuate existing issues in mental health (e.g., increase stigmatisation or discrimination) [65,77,83]	HP	Especially those who are medication naïve [68]
	B	Perception that PGxT is not cost-effective [68,76,81]	B	PGx can validate previous medication experiences [67,68,76,83,87,88]
2. Cognitive Participation				
2.1 Initiation	Ps	Some psychiatrists do not believe they are appropriate to drive implementation, in part due to:	B	Strong interest in adopting PGxT, due to belief it will yield patient, HCP, and system benefits [62,65,66,72,73]
	Ps	Lack of skills and knowledge relating to PGx [68,75,87]	HP	Belief that pharmacists are important in PGx implementation [62,68,90]
	Ps	Lack of confidence in PGx being effective [78]	HP	MDT approach to monitoring PGx outcomes is desired [86,90]
2.2 Enrolment	B	Perceived difficulty to integrate PGx into existing clinical pathways and services [69,70,88]	HP	Willingness from psychiatrists and prescribers to engage with other HCPs during implementation, particularly [68,86,90]:
			Ph	Pharmacists—perception that their role in PGx education, PGx patient counselling, PGx results interpretation and clinical service development will be key [68,90]
			HP	Genetic Counsellors—belief they can play a role in discussing PGx with patients [84,85]
			HP	Trust is key when HCPs collaborate on PGx [69,90]
2.3 Legitimation	Ps	Psychiatrist uncertainty about when to offer PGxT to patients—some psychiatrist sub-populations less likely to consider PGxT were: [66,75]	Ps	Psychiatrists believe PGx should be within their scope of practice [66,71,72]
	Ps	Older/more experienced psychiatrists [78]	Ps	Particularly offering PGx as an option to patients [66]
	Ps	Psychiatrists adopting a more psychosocial approach to mental health [75]	Ph	Pharmacists perceive PGx to be part of their clinical role expansion [90]
	Ps	Perception amongst psychiatrists that others in the profession lack the knowledge required to utilise PGxT [63]		
2.4 Activation	B	Ethical concerns about PGx tasks and activities [62,75,83,84]	HP	Agreement in tasks required to sustain PGx [68,72]
	HP	Gaining patient consent [62,72]	HP	Including obtaining consent [71,72], patient counselling [72], results interpretation, sharing results and updating results [68,69]
	B	Storage of genetic data and privacy issues [62,63,64,68,70,75,83,84]	HP	Belief that they can facilitate PGx-related activities [67,68,86]
	Ps	Psychiatrist concern about capacity to counsel patients about PGx and depth of information to provide [75,88]	B	Educating patients is essential [65,86] and improves their perception of PGx [68,70,76,82]
	HP	Concern about updating PGx results—as new evidence emerges [69,83]		
3. Collective Action				
3.1 Interactional Workability	HP	PGx may create extra burden during prescribing [67,68,69,75]		
	HP	Due to additional time required to complete PGx-related tasks [67,68,69,72]		
	B	Delay in obtaining PGx results can be problematic [67,68,70,74,75,83,88]	HP	PGx can help make prescribing decisions easier when PGx is delivered in a timely and accessible manner [74]
	HP	If PGx reports are too long or detailed [68]	B	PGx can make it easier to build patient rapport [66,67,83]
	Ps	If psychiatrists need to access support to utilise PGx results [63,67]		
	B	PGx may hinder rapport-building with patients [62,63,75]		
	B	Risk of misinterpretation of PGx results, particularly traffic-light systems of reporting that may oversimplify decision-making [68,76,88]		
3.2 Relational Integration	B	Lack of confidence in PGxT as an intervention, due to doubts over: [65,68,76,83,85]	HP	Perception that PGxT is safe and reliable helped contribute to confidence in PGxT as an intervention [62,68,72,75]
	HP	Ability of PGx to improve clinical outcomes [65,68,85]	HP	A belief that PGxT is a promising strategy to prescribing that will change practice and become a normal part of prescribing [62,66,71,72,75,85,87,89,90]
	Ps	Likelihood of PGx to influencing prescribing decisions [75,85]		
	B	Cost of conducting PGxT [62,63,66,67,68,72,75,76,81,83,84,87]		
3.3 Skillset Workability	HP	HCP knowledge and expertise gap for PGx in psychiatry [66,68,75,81,87]	HP	Relevant training and experience helps HCPs to be sufficiently knowledgeable and confident to use PGx [65,66,72,80]
	Ps	Psychiatrists lack awareness of the guidance for using PGxT in psychiatry [62,63,65,66,75,87,88]	B	Stakeholders are enthusiastic to learn more about PGx [62,66,70,75]
	Ps	Psychiatrists did not feel informed enough to identify when to offer PGxT or utilise PGxT results [66]	HP	Through e-learning, case studies, lectures and formal qualifications [62,66]
	Ps	Patient perception that HCPs may lack expertise to offer and use PGxT [83]	Ph	Pharmacists are confident in identifying when to offer PGx [66] and provide PGx counselling [90]
			Pa	Depressed patients have the psychological capacity to deal with PGx results [77]
			B	PGx education informs patients and manages expectations [76]
3.4 Contextual Integration	HP	Perceived lack of internal and external policy on PGx [71]	HP	Belief that psychiatry specific, practical PGx guidelines would help implementation [65]
	HP	Belief that there is a lack of basic and specialised education and training about PGx [62,63,68,72,75,90]	HP	Perception that national and international policy on PGx would help implementation [63,65,86]
	HP	A notion there is a lack of guidance and resources available to implement PGx [62,63,65,71,87]	HP	Staff and leadership engagement helped local implementation [80,86]
	Pa	Patients perceived to have a lack of education and awareness about PGx [70,73,74,82]	HP	Adoption of local PGx champions helps implementation [80,86]
4. Reflexive Monitoring				
4.1 Systemisation				
4.2 Individual Appraisal			HP	HCPs used their own clinical judgement to appraise PGx outcomes [90]
4.3 Communal Appraisal				
4.4 Reconfiguration			HP	Preference to adapt PGx for specific contexts, based on experience and feedback [88]

Description: Displayed in the table are the constructed barriers and facilitators mapped onto different constructs and sub-constructs of the NPT framework, and which stakeholder group the barrier or facilitator was constructed from: HP = healthcare professional, Pa = patient, B = both, Ps = psychiatrist and Ph = pharmacist. Sub-themes start with bullet points. ⇩ = decreased; ⇧ = increased.

**Table 6 jpm-14-01032-t006:** Summary of recommendations for policy, practice, and research.

Policy and Practice	Research
An implementation strategy for integrating PGx in mental health settings. Local guidance for the use of PGx in psychiatric settings. Workforce development plan to upskill and equip existing professionals with the skills and knowledge to deliver PGx services. Identify local PGx champions within organisations to act as ‘knowledge brokers’.	Use of implementation theory, like NPT, within future clinical PGx studies. Process evaluation of PGx implementation in mental health using a theoretical framework such as NPT. Exploration of attitudes towards using PGx in a broader range of mental illness, beyond depression. Further research to determine the role of pharmacy in PGx service development, implementation, and delivery. Evaluation of PGx cost-effectiveness in a wider range of healthcare systems.

Description: Displayed in the table is a summary of the key recommendations for policy, practice, and research resulting from the review findings.

## Data Availability

No new data were created or analysed in this study. Data sharing is not applicable to this article.

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
