# Peer review of "Normalising the Implementation of Pharmacogenomic (PGx) Testing in Adult Mental Health Settings: A Theory-Based Systematic Review"

_jpm, 2024, doi:10.3390/jpm14101032_

Round 1

Reviewer 1 Report

Comments and Suggestions for Authors

Title: settings? Amend does not seem right + A Theory-Based? All reviews are theory based

Abstract:  PGx firs use need to be full. define PGx early in the introduction for readers unfamiliar with the term.

Line 39 strengthen arguments by using GBD studies the 1:4 ppl argument is weak.

Line 42: which psychotropic med list class and supporting evidence

Keyword: +precision medicine

Noise can be reduced if terms unified e.g. mental health illness = psychiatric disorders and followed by psychiatric treatment

Line 55 95% of people possess genetic variants affecting drug response?? which genes or variants are most relevant to psychiatric medications.

Line 84-85 argument does not make sense. ?? Despite limited use of PGx in other specialties, the lack of PGx-informed psychiatry

Line 96 objectives can be removed

Line 201 QuADS better presented visually traffic light + summary

Line 221 Patient Public Involvement and Engagement (PPIE) is usually in original studies this call for question why IPD sys review and meta-analysis was not done

Table 1 add conclusions of each study/take away message. Although no formal effect size will be calculated I will good to present such information

Fig 2 add little information consider adding timeline of knowledge gained instead study descriptions. Surprisingly 2020-2022 was heavily active period >50% of our knowledge was over pandemic – can authors explain possible reasons?

Figure 3. refers to table 1 I think should be Table 4.

Discuss potential limitations of the study in greater detail, such as the diversity of the study settings. Heterogeneity is very large.

Discuss a balanced argument of the points related to the potential benefits of PGx in improving therapeutic outcomes vs the variability in clinical utility across different psychiatric conditions and medications. Also psychiatric conditions (and therefore meds) tend to coexist so thoughts about this?

PGx-informed prescribing is good pls provide detailed cost-benefit analyses or examples from healthcare systems where PGx has been implemented/tried. What technological requirements and infrastructure needed to support widespread PGx testing. Many psychiatric hospitals lack basic lab facilities and rely on near general hospitals. Thus existing regulatory guidelines for PGx testing in psychiatry.

Discuss ethics e.g. ethical considerations related to genetic testing, including patient consent, data privacy, and the potential for genetic discrimination. If it goes to national electronic record maybe it get abused too.

Provide balanced argument - Not all genetic tests have established clinical validity.

Burke W. Genetic tests: clinical validity and clinical utility. Curr Protoc Hum Genet. 2014 Apr 24;81:9.15.1-9.15.8. doi: 10.1002/0471142905.hg0915s81. PMID: 24763995; PMCID: PMC4084965.

Franceschini N, Frick A, Kopp JB. Genetic Testing in Clinical Settings. Am J Kidney Dis. 2018 Oct;72(4):569-581. doi: 10.1053/j.ajkd.2018.02.351. Epub 2018 Apr 11. PMID: 29655499; PMCID: PMC6153053.

Also discuss With advances in genomic sequencing technology, the number of reported gene-disease relationships has rapidly expanded. However, the evidence supporting these claims varies widely, confounding accurate evaluation of genomic variation in a clinical setting. 

Strande NT, Riggs ER, Buchanan AH, Ceyhan-Birsoy O, DiStefano M, Dwight SS, Goldstein J, Ghosh R, Seifert BA, Sneddon TP, Wright MW, Milko LV, Cherry JM, Giovanni MA, Murray MF, O'Daniel JM, Ramos EM, Santani AB, Scott AF, Plon SE, Rehm HL, Martin CL, Berg JS. Evaluating the Clinical Validity of Gene-Disease Associations: An Evidence-Based Framework Developed by the Clinical Genome Resource. Am J Hum Genet. 2017 Jun 1;100(6):895-906. doi: 10.1016/j.ajhg.2017.04.015. Epub 2017 May 25. PMID: 28552198; PMCID: PMC5473734.

Demkow U, Wolańczyk T. Genetic tests in major psychiatric disorders-integrating molecular medicine with clinical psychiatry-why is it so difficult? Transl Psychiatry. 2017 Jun 13;7(6):e1151. doi: 10.1038/tp.2017.106. PMID: 28608853; PMCID: PMC5537634.

Note: None of the above studies are my own

Reviewer 2 Report

Comments and Suggestions for Authors

Title of the Manuscript:

Normalising the Implementation of Pharmacogenomic (PGx) Testing in Mental Health Settings: A Theory-Based Systematic Review.

Manuscript ID: jpm-3184826

Comments:

The manuscript titled  “Normalising the Implementation of Pharmacogenomic (PGx) Testing in Mental Health Settings: A Theory-Based Systematic Review” seems to be exhaustive  and comprehensive and of  relevance to the field of mental health for implementing pharmacogenomic testing.

 The literature search performed on standard method, and has relevant recent literature cited. The number of cited literature is sufficient as well as relevant to the field.

 The statements and conclusions drawn are sufficiently supported by the referred citations.

 The figures and tables are sufficient to represent the data.

Minor points:

1.      Line 43: Please check the reference 2 for its appropriate placement.

2.      Line 52-54: Please rephrase the sentence for its clarity.

3.      Line 65: “A range of gene variants”…authors are requested to give example of some genes here.

4.      Lines 430-431: Please rephrase the sentence for more clarity.

5.      The annexure may be given as supplementary data.

Reviewer 3 Report

Comments and Suggestions for Authors

Reviewer Comments:

General Comments:

The manuscript is well-structured, but the English language requires minor improvements to enhance consistency and flow. I suggest revising some of the longer, complex sentences to make them shorter and more concise, which would improve readability.

Abstract:

The abstract provides a good overview of the study. However, I recommend briefly mentioning the potential impact of the findings on clinical practice and patient outcomes to make the significance of the research more apparent to the reader.

Introduction:

While the introduction is informative, it would benefit from a clearer articulation of the research gap the study aims to address. Additionally, I suggest setting more specific objectives that the study seeks to achieve.

Please explain how the normalization process is particularly suited for analyzing the implementation of pharmacogenomics (PGx). This would help clarify the rationale behind the chosen methodological approach.

Methods:

The detailed methodology is a strong point of the manuscript, ensuring that the systematic review is comprehensive, transparent, and rigorously conducted. However, consider setting out the Population, Intervention, Outcomes, Comparators, and Study design (PIOCS) criteria more explicitly to guide the reader through the process.

Results:

The narrative in the tables is currently difficult to read. Please adjust the text for clarity and consistency.

There is excessive detail in the paragraphs describing the results. Summarizing key findings more concisely would make the section more accessible and allow the reader to focus on the most important data.

Comments on the Quality of English Language

English language requires minor improvements.

Reviewer 4 Report

Comments and Suggestions for Authors

The manuscript “Normalising the Implementation of Pharmacogenomic (PGx) Testing in Mental Health Settings: A Theory-Based Systematic Review” analyses the main barriers and facilitators influencing a wide use of PGx in mental health settings, but mainly depression is the main topic. Description of the methodology is deep enough. Results are in acordance to recruited studies. Authors recognize limitations of their manuscript.

There are some aspects that could be improved in the paper:

1)      Information for implementation of pharmacogenetics in clinical practice can not only be obtained from FDA. There are more resources to check and help its implementation for practitioners such as PharmGKB database (that includes FDA information but also other agencies) or CPIC guidelines.

2)      It is justified that studies exploring PGx implementation in CAMH settings are excluded for this review. For that reason, title of the manuscript and/or aims of the study should specify that it is only for adults.

3)      In relation to figure 1, criteria for discarding studies (right squared information) could follow the same order for the different steps for consistency reasons.

4)      Typing mistake: PRISMA guidance instead of PRIMSA guidance (title of figure 1).

5)      Since the number of HCP (n=20), and even if it is included the number of HCPs and patients (n=2), does not match with the sum of psychiatrists (n=10), mixed professionals (n=7) and pharmacists (n=1), which other health professionals are included in the selected studies? Whoever they are, this situation should be highlighted as any other HCPs to describe the analysed population.

6)      In the number or HCP (lines 249-250) authors state that there is only 1 study with information from pharmacists but in the table there are more than one. Please, check the numbers.

7)      Among the included studies there are some of them that are quite old (more than 15-20 years ago). Could authors discuss the influence of such studies in the final results of the manuscript? Do more recent studies have overcome the limitations of the older ones or hold the same ideas?

8)      In table 3 it seems to follow alphabetical order but the last study. Please, reorder them consistently.

9)      In figure 3, please define what E +T means. Education and training are explained later on.

10)  I do not know if “time” is another barrier since it appears several times because of the ideas showing concern about delay in obtaining PGx results. Please, reconsider if appropiate.

11)  Could authors discuss if there are differences between general public opinion (patients included) and HCPs?

12)  Pharmacists are not usually prescribers in many countries neither genetic counsellors and it should be highlighted as a limitation in case of including pharmacist among HCP implementing PGx.

13)  Difference between pharmacogenetics and pharmacogenomics is not specified throughout the paper and many times it is using PGx term instead of the correct one. Please, clarify what PGx term includes.

Round 2

Reviewer 1 Report

Comments and Suggestions for Authors

None

Reviewer 2 Report

Comments and Suggestions for Authors

I want to thank the manuscript's authors for satisfactorily addressing all the concerns. 

Reviewer 3 Report

Comments and Suggestions for Authors

The manuscript has been improved and accepted for publication in its current form.

Reviewer 4 Report

Comments and Suggestions for Authors

The authors have adequately addressed my comments.